# Evaluation of Food Offerings for Workers in Commercial Foodservices from the Perspective of Healthiness and Sustainability

**DOI:** 10.3390/ijerph23010071

**Published:** 2026-01-02

**Authors:** Thaís de Gois Santos Marinho, Maria Luísa Meira Faustino, Maria Izabel de Oliveira Silva, Tatiane de Gois Santos, Ingrid Wilza Leal Bezerra, Priscilla Moura Rolim

**Affiliations:** 1Department of Nutrition, Federal University of Rio Grande do Norte (UFRN), Natal 59078-900, RN, Brazil; tatiane.gois.092@ufrn.edu.br (T.d.G.S.M.); luisa.meira.017@ufrn.br (M.L.M.F.); izabel.silva.070@ufrn.edu.br (M.I.d.O.S.); ingrid.bezerra@ufrn.br (I.W.L.B.); 2Department of Mechanical Engineering, Federal University of Rio Grande do Norte (UFRN), Natal 59078-900, RN, Brazil; thais.gois@ufrn.br

**Keywords:** food services, menus, workers, food security, environmental sustainability

## Abstract

**Highlights:**

**Public health relevance—How does this work relate to a public health issue?**
Examines how food insecurity, excess weight, and meal quality intersect among workers who work in food services, a population often overlooked in public health nutrition.Links workers’ social vulnerability, nutritional status, and menu healthiness/sustainability to broader public health challenges, including non-communicable diseases and inequities in access to adequate food.

**Public health significance—Why is this work of significance to public health?**
High food insecurity and overweight/obesity among workers, alongside menus high in lipids, sodium, and environmental impact, suggesting health risks and inequities.By integrating social, nutritional, and environmental indicators, the study advances understanding of how workplace food environments can support or undermine health.

**Public health implications—What are the key implications or messages for practitioners, policy makers and/or researchers in public health?**
Urges practitioners to redesign menus and procurement to reduce sodium and lipids, prioritize low-impact proteins, and integrate food security and sustainability.Highlights for researchers the importance of multidimensional approaches for scalable monitoring tools and interventions.

**Abstract:**

Aims: To evaluate the quality of lunch menus for workers in commercial food services across social, health and environmental sustainability dimensions. Methods: Mixed methods were applied to five restaurants. Data collection included a sociodemographic questionnaire, the Brazilian Food Insecurity Scale (EBIA), workers’ nutritional status, nutritional composition of 111 lunch menus, and environmental footprints. Data triangulation integrated caloric–nutritional adequacy, food insecurity, obesity, protein supply, and environmental footprints. Results: We assessed 261 participants (71.6% male; average age 32.3; 53.5% with a high school education). Food insecurity affected 53.3% and was associated with income, education, household composition, and municipality (*p* < 0.05). Nutritional status (*n* = 438) showed 68.3% were overweight/obese; obesity affected 42.7% of women and 30.5% of men. Menu analyses (*n* = 111) showed adequate energy and protein, but excessive lipids and sodium and reduced carbohydrates. Environmental analyses indicated beef had the highest impact; protein type was more influential than quantity, indicating no simple linear nutrition–impact relationship. Conclusion: Widespread food insecurity and obesity co-occurred with menus characterized by excessive lipids, sodium, and beef-driven impacts. The findings highlight that health and sustainability outcomes depend on both menu quality and social context, necessitating integrated, multidimensional policies.

## 1. Introduction

The One Health concept sustainably integrates the health of people, animals, and ecosystems, with food security as a central pillar [1,2]. In this context, ensuring the right to healthy eating through sustainable practices is essential for environments that promote health and well-being [3]. This perspective aligns with the FAO concept of sustainable diets, which combines health, low environmental impact, accessibility, safety, and cultural acceptance [4], as well as the sustainable nutrition framework proposed by Von Koerber et al., which emphasizes minimally processed foods and consideration of the whole food chain [5].

Food services, regulated by CFN Resolution N°. 600/2018 [6], play an important role in providing safe and adequate meals, directly influencing workers’ health and eating habits [7]. When diets are inadequate, the Human Right to Adequate Food (HRAF) may be compromised, potentially leading to food insecurity, which can progressively affect individuals and communities. Food insecurity can be measured using the Brazilian Food Insecurity Scale (EBIA), which assesses population groups’ perceptions of their eating practices [8].

Grounded in the Brazilian Organic Law on Food and Nutrition Security (LOSAN) [9], food and nutrition security involve regular access to sufficient, quality food in a culturally, socially, economically, and environmentally sustainable manner. A balanced diet is essential for improving worker health by helping prevent non-communicable diseases, improving nutritional status, and supporting performance [10]. Despite the sector’s expansion, food service workers may experience precarized working conditions, including exhausting working conditions, low wages, high turnover, and a high prevalence of overweight and obesity [11].

Food insecurity remains a major public health challenge worldwide, reflecting persistent barriers to regular access to sufficient, safe, and nutritious food. Global monitoring reports continue to show high prevalence of moderate or severe food insecurity and persistent hunger, with disproportionate impacts on socially vulnerable groups and marked territorial inequalities [12].

In Latin America and the Caribbean, recent assessments indicate improvements in hunger and food insecurity in some countries; however, progress remains uneven and vulnerable populations continue to face structural barriers to healthy diets, which remain costly and constrained by socioeconomic conditions and climate-related instability [13].

In Brazil, national household survey data based on the Brazilian Food Insecurity Scale (EBIA) indicate a recent reduction in the proportion of households experiencing some degree of food insecurity between 2023 and 2024, although a substantial share of households remains affected [14].

Within this context, workers in commercial food services are a relevant population because they are directly exposed to workplace food environments and may face socioeconomic constraints and demanding working conditions. Reliance on meals provided at work, together with household food insecurity, may influence dietary patterns and health, supporting the need for multidimensional assessments [11].

Sustainability is increasingly linked to nutrition and menu planning, particularly in commercial food services and in the role of nutritionists [15]. Sustainable practices can enhance service quality and generate social and environmental benefits [16], and they are often assessed using indicators such as environmental footprints, which are key for healthy and sustainable diets [17]. However, the scarcity of studies integrating sustainability, collective food services, and food security limits understanding of these challenges and highlights the need for systemic research to inform policies and practices in the sector.

By integrating social, nutritional, and environmental indicators within the same institutional setting, this study provides a multidimensional assessment of commercial food services, which can inform menu planning and workplace strategies aimed at healthier and more sustainable food environments.

Thus, this study aimed to evaluate the quality of food provided to workers in commercial food services across three dimensions: health, social, and environmental. Specifically, it sought to answer: how do the social, health, and environmental aspects of meals provided by these services influence workers’ health, food security, and engagement with sustainability within the institutional context? This paper is organized as follows: Methods describes the study design and analyses; Results presents the findings; and Discussion interprets the results and their implications for practice and policy.

## 2. Materials and Methods

### 2.1. Study Design and Setting

This study was conducted in five commercial food service restaurants in Natal, Rio Grande do Norte, in Northeast Brazil. To preserve confidentiality, establishments were coded as R1–R5 and selected by convenience based on institutional access, authorization, and the availability of the required records. Figure 1 shows the location of Natal (Rio Grande do Norte) in Northeast Brazil.

### 2.2. Participants and Sampling

All workers from the five restaurants were invited to participate through convenience sampling with voluntary enrolment. During the study period, the five establishments employed approximately 600 workers (an exact number per unit was not available). Participation was independent across study components and occurred after reading and signing the informed consent form (ICF). Accordingly, 261 participants completed the sociodemographic questionnaire and the short EBIA (lower adherence due to time demands and sensitive information), 438 underwent anthropometric assessment (higher adherence because it was brief and minimally invasive), and lunch menus from all restaurants were collected and analyzed over four consecutive weeks (*n* = 111) using establishment records.

### 2.3. Data Collection Components

Data collection was organized into three study components: social (sociodemographic profile, eating practices, and food insecurity assessed by EBIA), health (anthropometric assessment and nutritional/processing characteristics of lunch menus), and environmental (carbon, water, and ecological footprints of the menus).

### 2.4. Social Component: Questionnaire and Food Insecurity

To assess the sociodemographic profile and eating practices, a questionnaire was administered online (Google Forms) and in print. Data collected included sex, age, race/ethnicity, residence, education, income, household size, length of employment, sector, and job role. Eating practices included questions on consuming meals at home and the worker’s responsibility for food purchasing and preparation. Food insecurity was measured using the short version of the EBIA, composed of five yes/no questions regarding household food conditions in the three months prior to data collection. Cut-off points followed Coelho et al. [18]: food security (0 points), mild food insecurity (1–2), moderate food insecurity (3–4), and severe food insecurity (5).

### 2.5. Health Component: Anthropometry and Nutritional Assessment

Workers’ nutritional status was assessed using Body Mass Index (BMI), based on weight (digital scale) and height (stadiometer, 150 cm, 0.1 cm precision), following the criteria of the Brazilian Food and Nutrition Surveillance System (SISVAN) [19] and the World Health Organization (WHO) [20] cut-off points. Average Total Energy Value (TEV) was estimated from Total Energy Expenditure (TEE). Basal Metabolic Rate (BMR) was calculated according to FAO/WHO [21], assuming a light Physical Activity Level (PAL). Nutritional analyses of menus were based on recipe standardized sheets, considering per capita amounts of foods consumed at lunch over one month. Nutrient composition (macronutrients, iron, sodium [22], vitamins A and C, and fiber) was obtained from the Brazilian Food Composition Table [23] and complemented by other databases, with calculations based on net weight (g) of each ingredient. Nutritional recommendations followed the International Life Sciences Institute Brazil (ILSI) [24], and Dietary Reference Intakes (DRI) [25]; calcium followed [26].

### 2.6. Menu Composition and Food Processing Assessment

Food purchase records (September 2024) and suppliers’ invoices were used to assess food acquisition, level of processing, and critical ingredients. Supplier origin was determined based on invoice records, categorized by distance to the meal preparation site [27,28,29]: local (≤164 km), state (Rio Grande do Norte), regional (Northeast), national (other Brazilian regions), and international (other countries). Food classification followed Martinelli et al. [30], grouped into 11 food categories, and then subdivided by processing level (based on Brazilian Dietary Guidelines [31] and NOVA classification [32]), and the presence of critical ingredients [33,34,35] and trans fats [36] in packaged foods. Nutrient profile assessment followed the Pan American Health Organization (PAHO) model, with thresholds of total fat ≥ 30% total energy, saturated fat ≥10% total energy, trans fat ≥1% total energy, and sodium ≥ 1 mg/kcal [36,37].

### 2.7. Environmental Component: Footprints

The environmental component assessed the environmental impact of the menus by evaluating carbon (CF), water (WF), and ecological (EF) footprints. The methodology followed Garzillo et al. [38] and Garzillo et al. [39]. The analysis was based on lunch menu records offered over four consecutive weeks in each restaurant, with mean weekly and monthly food quantities calculated for analysis. Footprint coefficients were derived from the publication “Footprints of foods and culinary preparations consumed in Brazil” [39].

### 2.8. Statistical Analysis and Data Triangulation

Data collected via Google Forms were entered into Excel, coded, and analyzed in Jamovi (v.2.3). Data were described as frequencies and tested for normality (Shapiro–Wilk test). Environmental footprints were compared with the Kruskal–Wallis test, followed by the Dwass–Steel–Critchlow–Fligner post hoc test. Results were expressed as means and standard deviations or as medians and quartiles (Q1–Q3). Associations between sociodemographic variables and perception of food insecurity were examined using the Chi-square test, adjusted for control variables. Statistical significance was set at *p* < 0.05. Data triangulation was applied to strengthen analytical robustness and minimize bias, integrating social, health, and environmental information, based on the framework of Zanin et al. [40]. Two triangulation streams were defined: (I) caloric-nutritional adequacy of menus × obesity × food insecurity, and (II) protein supply of menus × carbon footprint × water footprint.

### 2.9. Ethics

The study received approval from the institutional research ethics committee, and all procedures adhered to the Declaration of Helsinki. Written informed consent was obtained as required.

## 3. Results

To directly address the research question, results are presented in three subsections: (i) the influence of served meals on worker health; (ii) the influence of served meals on food security; and (iii) sustainability-related findings within the institutional context. An integrated synthesis using data triangulation is then provided to summarize multidimensional patterns across the five restaurants.

### 3.1. Influence of Served Meals on Food Security

A total of 261 workers were evaluated (mean age 32.3 years). Most were male (71.6%) and self-identified as Black or Brown (63.5%). The majority lived in Natal (70.1%) and shared their household with 1–3 people (77.3%). In terms of education, 53.5% had completed high school, and 71.8% were married. Most households reported a family income between 1 and 2 minimum wages. Length of employment was 0–1 year (41%), 2–5 years (30%), and >5 years (29%). Regarding work organization, 45.2% were in operational sectors, 33.2% in customer service, 18.1% in support functions, and 3.3% were apprentices. With respect to eating practices, 70.2% of workers were responsible for purchasing food and 70.6% for meal preparation.

As shown in Table 1, 53.3% of participants presented some level of food insecurity (FI): 31.8% mild, 10.7% moderate, and 10.7% severe. EBIA scores ranged from 0 to 5, with a mean of 1.34 (SD = 1.69). Association analyses of dichotomized variables (α = 0.05) indicated statistically significant relationships (*p* < 0.05) with municipality of residence, household composition, monthly income, education level, and job type, indicating that these factors were associated with FI (see Table 1 for exact *p*-values).

The synthesis of aggregated indicators across the five evaluated units, presented in Table 2, supported a comparative qualitative analysis and highlighted three findings: (1) higher sodium and lipid contents were strongly associated with obesity (R2 and R5); (2) food security influenced but did not solely explain nutritional outcomes, as R5 showed high food security but elevated obesity despite broad macronutrient availability; and (3) environmental footprints were not directly related to clinical outcomes, as favorable health profiles with higher impacts (R3) and poorer indicators with lower impacts (R2) were both observed.

### 3.2. Influence of Served Meals on Workers’ Health

The analysis of workers’ nutritional profiles assessed the nutritional status of 438 workers (77.6% male, *n* = 340; 22.4% female, *n* = 98). Mean anthropometric measurements were: weight = 79.3 kg (SD = 15.3), height = 1.69 m (SD = 0.086), and BMI = 27.87 kg/m^2^ (SD = 5.00). Overall, 68.3% of workers were overweight or obese; 30.4% were within the normal weight range, and 1.4% were underweight. Analysis by sex revealed significant differences in obesity prevalence, which was higher among women (42.7%, *n* = 41) compared with men (30.5%, *n* = 104).

Monthly average caloric content and adequacy for R1 to R5: R1—876 kcal (97% adequacy); R2—965 kcal (112%); R3—1003 kcal (113%); R4—909 kcal (101%); and R5—1329 kcal (160%). Macronutrient assessment showed carbohydrate values (35.6% to 41.8% of TEV) were below the minimum recommended level of 45% [24]. Conversely, lipid values (36.8% to 41.4% of TEV) exceeded the upper limit of 35% [32]. Protein levels (20.6% to 24.5%) remained adequate. Micronutrient evaluation indicated iron and vitamin C were adequate, but calcium and sodium levels were inadequate in all establishments; vitamin A inadequacy was observed only in R3 and R4.

Findings on food procurement and processing indicated that 54.1% of purchases originated from local suppliers, 43.2% from state-level suppliers, and 2.7% from national suppliers. According to the (PAHO) nutrient profile analyses [36], 24.7% of processed foods contained high amounts of critical ingredients. Based on the NOVA classification [32], 59.8% of items were unprocessed or minimally processed; ultra-processed foods accounted for 18.6%, processed foods for 7.2%, and processed culinary ingredients for 14.4%. In the monthly categorization (97 items), vegetables were the most representative group (21.6%); processed foods accounted for 6.2%, and sauces high in sodium/fat represented 3.1%.

### 3.3. Sustainability-Related Findings Within the INSTITUTIONAL Context

Sustainability-related findings derive from an analysis conducted via a weekly survey of ingredients used in five restaurants over one month. Each item was linked to estimates of carbon footprint (CF; g CO_2_e), water footprint (WF; liters), and ecological footprint (EF; g·m^2^) [39]. Calculations were performed using the original units of Garzillo’s coefficients (g CO_2_e and g·m^2^); for reporting and comparison with “climate budget” recommendations, values were converted to kilograms (1 kg = 1000 g), while the water footprint was kept in liters.

Data were grouped into eight food categories. Normality was assessed using the Shapiro–Wilk test. Given the non-normal distribution (*p* < 0.05), non-parametric Kruskal–Wallis tests were applied to identify differences among food groups (Table 3). Table 2 summarizes average environmental footprints as follows: carbon—2.3 kg CO_2_e, water—1.94 L, and ecological—12.5 kg/m^2^. To contextualize these values, WWF [41], recommends a maximum “climate budget” of 0.5 kg CO_2_e per main meal (lunch or dinner) or 1.6 kg CO_2_e per person per day. Using the categories Low (0.1–0.5 kg CO_2_e), Medium (0.5–1.5 kg CO_2_e), and High (>1.6 kg CO_2_e), the average impact of the evaluated restaurants is classified as High.

The Kruskal–Wallis test indicated significant differences among food groups for all environmental footprints (*p* < 0.001 in all cases). Post hoc Dwass–Steel–Critchlow–Fligner tests (Table 3) further showed that beef had the highest environmental impact, whereas fruits and vegetables/cereals/legumes exhibited the lowest impacts.

### 3.4. Data Triangulation Across Dimensions

The qualitative analysis through data triangulation (Figure 2), comparing menu adequacy, obesity prevalence, and food insecurity (FI), revealed no simple linear relationship between menu adequacy and obesity. R5 presented very high adequacy (~160%) and elevated obesity (~40%) despite lower FI (40%). R2, with intermediate adequacy (~112%), had the highest obesity (~44%) and high FI (60%). In contrast, R3 had similar adequacy (~113%) but lower obesity (~27%) and moderate FI (43%). R1 combined lower adequacy (~97%) and very high FI (65%) with intermediate-to-low obesity (~29%). These contrasts indicate that variation in obesity cannot be explained solely by dietary adequacy or FI.

Figure 3, extending the triangulation, presents a greyscale scatter plot relating the carbon and water footprints for R1–R5. Each point is identified by a distinct symbol, with marker shade encoding the menu’s protein content (≈50–72 g). This greyscale scheme with distinct symbols ensures unit differentiation and print legibility. A monotonic relationship was observed: units with higher carbon footprints also exhibited greater water use. R4 appeared at the upper extreme, followed by R3; R2 showed the lowest footprints, R1 had similar values, and R5 occupied an intermediate position but with higher protein content. The analyses indicate that protein quantity alone does not explain environmental differences: R5, despite higher protein offerings, had a lower impact than R3 and R4. This suggests that protein type and side-dish composition exert a stronger influence on environmental outcomes. Notably, R2 combined low footprints with lower protein density and ingredients of lower environmental intensity.

## 4. Discussion

The primary findings of this study demonstrated a high prevalence of food insecurity (FI) among workers, especially those with lower education, lower income, and residents of peripheral municipalities. A high occurrence of excess weight (overweight/obesity), particularly among women, was also observed, along with nutritional inadequacies in the menus and potential negative environmental impacts. The following discussion explores these results in light of current literature, relating them to social, economic, and dietary factors that influence health and sustainability in the workplace.

Sociodemographic conditions profoundly influenced the perception of FI. Higher prevalence was noted in peripheral municipalities (*p* < 0.05), underlining territorial inequalities. Larger households demonstrated greater FI (*p* < 0.05), likely due to increased food demand under constrained income. Monthly income exhibited a strong association (*p* < 0.05): rates were highest among individuals earning up to R$ 2824.00, a finding consistent with literature identifying income as a key determinant of FI [42,43,44]. Regarding education (*p* < 0.05), lower educational attainment (primary or secondary schooling) was associated with higher vulnerability, restricting the full exercise of the Human Right to Adequate Food (HRAF) [42,44]. Workers in operational roles also experienced higher levels of FI (*p* < 0.05) (see Table 1 for exact *p*-values).

In contrast, age, sex, marital status, race/ethnicity, length of employment, and work sector showed no statistically significant association with FI (*p* > 0.05). Similar findings were reported by Fidelis et al. [42] and Falcão et al. [43], who observed food insecurity prevalences of 44.4% and 53.7% among food handlers, mainly associated with income and education. In this context, previous studies emphasize that low educational level, reduced income, and larger household size are key factors contributing to household food insecurity [43,45]. Work organization factors (e.g., shiftwork and atypical schedules) may further exacerbate vulnerability to food insecurity, reinforcing that this outcome is not explained by income and education alone [46]. Given this scenario, ensuring access to healthy and adequate food for these workers is essential, as it contributes not only to improved quality of life but also to greater productivity and efficiency in the service sector [45].

Another relevant finding was the higher prevalence of excess weight (overweight/obesity) among women, consistent with results from studies conducted in restaurants and institutional kitchens [47]. In contrast, Costa et al. [48] reported higher obesity prevalence and cardiometabolic risk among men. These findings suggest an increased risk of cardiovascular disease, possibly linked to unhealthy eating habits [49]. In menu assessment, macronutrient levels were similar to those reported by Andrade et al. [50], who also found comparable carbohydrate and lipid contents in an institutional food service. However, unlike the present study, their protein levels were higher. Regarding micronutrients, calcium inadequacy was observed across all restaurants due to the low presence of calcium-rich foods at lunch; sodium excess was consistent with previous studies [51,52] and vitamin A deficiency was identified in Restaurants R3 and R4. Conversely, fiber intake was adequate, confirming findings from similar research [51,52].

Regarding food group categorization, the four most representative categories (vegetables; meats, eggs, and derivatives; fruits and juices; and cereals, breads, and pasta) consisted mainly of unprocessed or minimally processed foods, which form the basis of a healthy diet according to the Brazilian Dietary Guidelines [31]. Concerning environmental footprints, several studies support these findings, indicating that the foods contributing most to environmental impacts, particularly the carbon footprint, are red meats, as well as pork, poultry, and dairy products [53,54,55].

The results showed that restaurants offering higher protein levels had greater environmental footprints, indicating a direct relationship between elevated protein consumption and increased meal impacts, whereas those with lower impacts also had reduced protein and calorie offerings. However, Restaurant R5 deviated from this pattern: despite providing the highest protein content, it did not present the largest environmental footprints. This outcome can be explained by the predominance of lower-impact protein sources, such as fish (three times) and poultry or eggs (seven times) throughout the week.

Similarly to the study by Pörtner et al. [55], which integrated nutritional quality and environmental footprint assessments of food services in hospitals and long-term care institutions, the present study also adopts a comprehensive approach to evaluate menu quality and its impacts. The use of nutritional and environmental indicators helps identify critical nutrients, such as excess lipids and sodium, and highlight priority areas for improving dietary quality and sustainability. The analyses of environmental footprints (carbon, water, and ecological) further reveals the main sources of impact and supports more balanced strategies for menu planning in food service operations.

Data triangulation showed no linear relationship among nutrition, food insecurity, obesity, and environmental impacts: similar menus can produce different outcomes depending on the social context, and higher protein content does not necessarily result in greater footprints, depending on food sources. These findings highlight the need for integrated strategies that simultaneously address nutritional, social, and environmental dimensions, guiding policies and practices toward healthier, more equitable, and sustainable food environments.

However, this study has limitations, including a convenience sample, restriction to five restaurants in a single city, and exclusive use of BMI. Convenience sampling may introduce selection bias and limit representativeness, which reduces the generalizability of the findings to other commercial food services and regional contexts. Restriction to a single city also constrains external validity, as social vulnerability, work organization, and food provision patterns may vary across territories. In addition, BMI does not capture body composition or fat distribution and may under or overestimate cardiometabolic risk when used as the sole anthropometric indicator.

Future research should expand this assessment to different cities and regions, using larger samples and designs that improve comparability across units. Incorporating complementary anthropometric measures (e.g., waist circumference and/or body composition) and individual dietary assessments would help clarify mechanisms linking food provision, food insecurity, and excess weight. Longitudinal and/or intervention studies could also test menu and management changes (such as reducing sodium and lipids and shifting protein choices) to support healthier and more sustainable workplace food environments.

## 5. Conclusions

This study aimed to evaluate the quality of food offered to workers in commercial food services by integrating health, social context, and environmental sustainability. The findings revealed weaknesses in the food offered, with a high prevalence of overweight and obesity, mainly among women, along with food insecurity associated with low income, larger households, and lower education levels. Nutritional inadequacies in carbohydrates, lipids, and sodium highlight challenges to health promotion, while the high consumption of meats, particularly beef, intensifies environmental impacts. Although unprocessed or minimally processed foods predominated, the presence of ultra-processed products and critical ingredients still compromises nutritional quality and sustainability. In practical terms, these findings support revising menu planning and procurement practices to reduce sodium and lipid density while expanding preparations with healthier nutritional profiles. They also suggest prioritizing lower-impact protein sources (e.g., reducing the share of beef where feasible) while maintaining nutritional adequacy. Finally, routine monitoring of nutrition and sustainability indicators can be implemented as a management tool in commercial food services.

The results reinforce the interdependence between health and sustainability in the food environment, as obesity is influenced not only by menu composition but also by social factors and working conditions. The origin and type of protein had a greater environmental influence than its quantity. Therefore, it is necessary to restructure food service practices, with nutritionists developing menus that integrate nutrition, sustainability, and food security. This recommendation aligns with evidence on healthy food service guidelines for worksites and institutional settings, which support the use of formal procurement/offer standards and food-environment strategies to improve the nutritional quality of foods served and facilitate healthier choices [56].

A systemic, evidence-based, and multidimensional approach is essential to strengthen public policies and institutional strategies aimed at promoting worker health and the sustainability of the food system.

## Figures and Tables

**Figure 1 ijerph-23-00071-f001:**
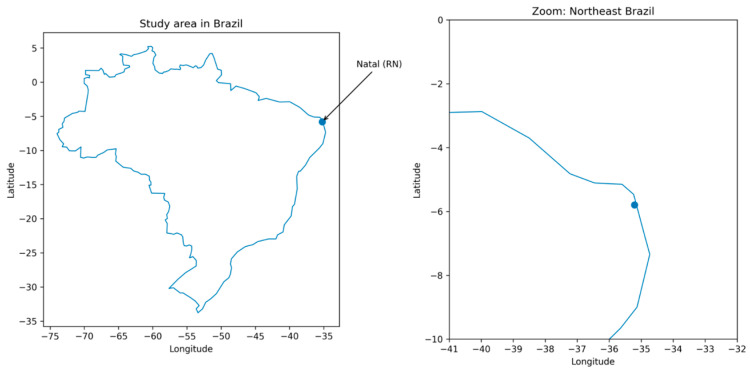
Study area location in Brazil. The dot indicates Natal (Rio Grande do Norte), Northeast Brazil, where the study was conducted.

**Figure 2 ijerph-23-00071-f002:**
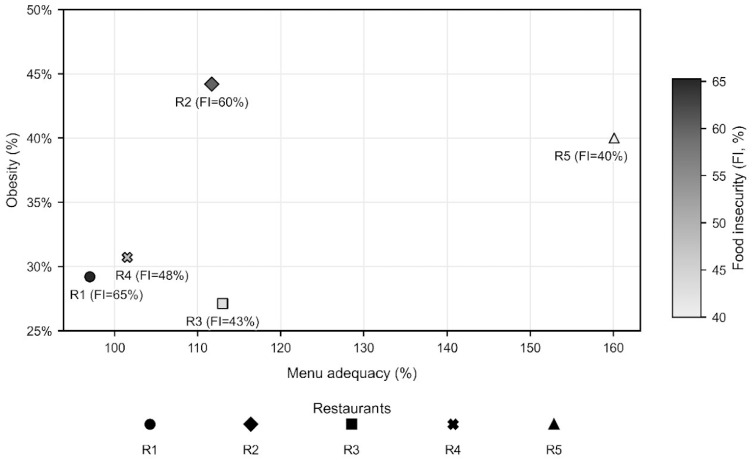
Triangulation of menu adequacy, obesity prevalence, and food insecurity in commercial restaurants.

**Figure 3 ijerph-23-00071-f003:**
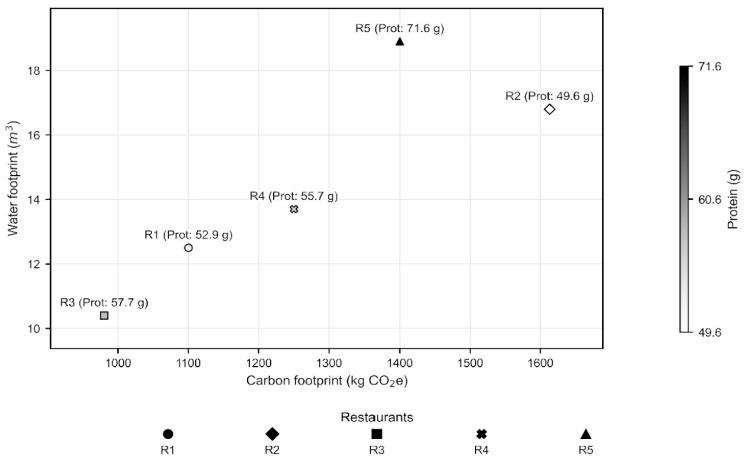
Triangulation of proteins offered in menus, carbon footprint, and water footprint of commercial restaurants.

**Table 1 ijerph-23-00071-t001:** Sociodemographic characteristics and food and nutrition security of workers in commercial food service establishments (*n* = 261).

	Food Insecurity—N (%)			Food Security—N (%)	Total	*p*-Value **
Age *	Mild FI:	Moderate FI:	Severe FI:			
<24	20 (32.3%)	9 (14.5%)	4 (6.4%)	29 (46.8%)	62	0.979
≥24	62 (31.3%)	19 (9.6%)	24 (12.1%)	93 (47.0%)	198	
Total	138 (53.1%)			122 (46.9%)	260 *	
Gender						
Female	20 (27.03%)	7 (9.46%)	7 (9.46%)	40 (54.1%)	74	0.136
Male	63 (33.69%)	21 (11.23%)	21 (11.23%)	82 (43.9%)	187	
Total	139 (53.3%)			122 (46.7%)	261	
Marital status						
With partner	41 (33.06%)	15 (12.1%)	12 (9.68%)	56 (45.2%)	124	0.626
Without partner	42 (30.66%)	13 (9.49%)	16 (11.68%)	66 (48.2%)	137	
Total	139 (53.3%)			122 (46.7%)	261	
Race/ethnicity						
White/Asian	27 (28.13%)	12 (12.5%)	7 (7.29%)	50 (52.1%)	96	0.187
Brown/Black/Indigenous	56 (33.94%)	16 (9.7%)	21 (12.73%)	72 (43.6%)	165	
Total	139 (53.3%)			122 (46.7%)	261	
Municipality of residence						
Other	19 (38.78%)	6 (12.24%)	8 (16.33%)	16 (32.7%)	49	0.028
Natal/Parnamirim-RN-Brazil	64 (30.19%)	22 (10.38%)	20 (9.43%)	106 (50.0%)	212	
Total	139 (53.3%)			122 (46.7%)	261	
Household size						
Up to 2 people	25 (28.41%)	3 (3.41%)	10 (11.36%)	50 (56.8%)	88	0.020
More than 2 people	58 (33.53%)	25 (14.45%)	18 (10.4%)	72 (41.6%)	173	
Total	139 (53.3%)			122 (46.7%)	261	
Monthly income *						
≤R$ 2824.00	44 (33.33%)	17 (12.88%)	21 (15.91%)	50 (37.9%)	132	0.001
>R$ 2824.00	28 (27.72%)	8 (7.92%)	5 (4.95%)	60 (59.4%)	101	
Total	123 (52.8%)			110 (47.2%)	233 *	
Education level *						
Elementary or high school	69 (33.01%)	25 (11.96%)	25 (11.96%)	90 (43.1%)	209	0.023
Higher education or postgraduate	14 (27.45%)	3 (5.88%)	3 (5.88%)	31 (60.8%)	51	
Total	139 (53.5%)			121 (46.5%)	260 *	
Job type						
Base position	53 (29.94%)	25 (14.12%)	22 (12.43%)	77 (43.5%)	177	0.128
Technical/superior	30 (35.71%)	3 (3.57%)	6 (7.14%)	45 (53.6%)	84	
Total	139 (53.3%)			122 (46.7%)	261	
Occupational sector						
Administrative	9 (31.03%)	1 (3.45%)	1 (3.45%)	18 (62.1%)	29	0.079
Operational	74 (31.9%)	27 (11.64%)	27 (11.64%)	104 (44.8%)	232	
Total	139 (53.3%)			122 (46.7%)	261	
Job position type						
Operational/base position	64 (33.33%)	22 (11.46%)	26 (13.54%)	80 (41.7%)	192	0.006
Technical/managerial position	19 (27.54%)	6 (8.7%)	2 (2.9%)	42 (60.9%)	69	
Total	139 (53.3%)			122 (46.7%)	261	
Responsible for purchasing food at home						
No	29 (35.37%)	7 (8.54%)	4 (4.88%)	42 (51.2%)	82	0.327
Yes	54 (30.17%)	21 (11.73%)	24 (13.41%)	80 (44.7%)	179	
Total	139 (53.3%)			122 (46.7%)	261	
Participates in meal preparation at home						
No	22 (31.88%)	4 (5.8%)	6 (8.7%)	37 (53.6%)	69	0.182
Yes	61 (31.77%)	24 (12.5%)	22 (11.46%)	85 (44.3%)	192	
Total	139 (53.3%)			122 (46.7%)	261	

Legend: * Missing data were excluded from analyses. *** p*-value from Pearson’s chi-square test. FI = Food Insecurity; FS = Food Security.

**Table 2 ijerph-23-00071-t002:** Multidimensional analysis of restaurants according to workers’ food and nutrition security and nutritional and sustainability parameters of menus.

Rest.	EBIA		Anthropometry		Menu Adequacy		Environmental Footprints (Monthly Mean)		
							CF kgCO_2_	WF m^3^	EF kgm^2^
R1	Food Insecurity Levels		BMI classification		Adequacy:	97%	1613	1483	7.6
	Severe FI:	13.7%	Underweight:	1.8%	Protein (g)	52.9			
	Mild FI:	36.8%	Normal weight:	38.1%	Lipids (g)	35.9			
	Moderate FI:	14.7%	Overweight:	31.0%	Carbohydrates (g)	85.4			
	FS:	34.7%	Obesity I–III:	29.2%	Sodium (mg)	1156			
R2	Food Insecurity Levels		BMI classification		Adequacy:	112%	1479	1385	6.0
	Severe FI:	13.3%	Underweight:	2.3%	Protein (g)	49.6			
	Mild FI:	40.0%	Normal weight:	18.6%	Lipids (g)	44.4			
	Moderate FI:	6.7%	Overweight:	34.9%	Carbohydrates (g)	85.9			
	FS:	40.0%	Obesity I–III:	44.2%	Sodium (mg)	1493			
R3	Food Insecurity Levels		BMI classification		Adequacy:	113%	2835	2458	14.0
	Severe FI:	4.5%	Underweight:	0.0%	Protein (g)	57.7			
	Mild FI:	29.2%	Normal weight:	39.0%	Lipids (g)	41.7			
	Moderate FI:	9.0%	Overweight:	33.9%	Carbohydrates (g)	99.1			
	FS:	57.3%	Obesity I–III:	27.1%	Sodium (mg)	950			
R4	Food Insecurity Levels		BMI classification		Adequacy:	102%	3411	2607	21.0
	Severe FI:	17.4%	Underweight:	1.6%	Protein (g)	55.7			
	Mild FI:	21.7%	Normal weight:	29.1%	Lipids (g)	37.6			
	Moderate FI:	8.7%	Overweight:	38.6%	Carbohydrates (g)	95.1			
	FS:	52.2%	Obesity I–III:	30.7%	Sodium (mg)	1487			
R5	Food Insecurity Levels		BMI classification		Adequacy:	160%	2170	1806	14.0
	Severe FI:	12.0%	Underweight:	1.1%	Protein (g)	71.6			
	Mild FI:	20.0%	Normal weight:	22.1%	Lipids (g)	58.8			
	Moderate FI:	8.0%	Overweight:	36.8%	Carbohydrates (g)	128.5			
	FS:	60.0%	Obesity I–III:	40.0%	Sodium (mg)	2242			
Average environmental footprints across restaurant							2302	1.95	12.52

Legend: FI = Food Insecurity; FS = Food Security; BMI = Body Mass Index; CF = Carbon Footprint; WF = Water Footprint; EF = Ecological Footprint; EBIA = Brazilian Food Insecurity Scale. The complete menu adequacy assessment is presented in the Appendix A, with detailed results.

**Table 3 ijerph-23-00071-t003:** Mean environmental footprints (carbon, water, and ecological) by food group, with grouping by statistical significance.

Food Group	CF (kg CO_2_e)	WF (m^3^)	EF (kg·m^2^)
Beef	3685.95 ^a^	2599.19 ^a^	12.72 ^a^
Poultry and eggs	573.45 ^b^	729.09 ^b^	0.43 ^ab^
Pork	814.66 ^b^	864.68 ^b^	0.33 ^b^
Fish and seafood	593.62 ^b^	14.10 ^bc^	25.83 ^b^
Dairy products	74.04 ^c^	54.88 ^c^	0.06 ^c^
Fruits and vegetables	28.02 ^d^	52.37 ^d^	0.02 ^d^
Cereals, legumes, and pasta	46.01 ^d^	70.61 ^d^	0.03 ^d^
Others	53.72 ^d^	48.66 ^d^	0.05 ^d^

Legend: Means followed by the same letter within a column do not differ significantly according to the Kruskal–Wallis test with the Dwass–Steel–Critchlow–Fligner post hoc test (*p* < 0.05). Letters indicate homogeneous groupings, summarizing detailed results provided in the Appendix A.

## Data Availability

The data presented in this study are not publicly available due to privacy and ethical restrictions, as they contain individual-level information about workers and institutional characteristics. The datasets are stored in a secure institutional drive. De-identified data may be made available from the corresponding author upon reasonable request and with permission from the participating institutions and the research ethics committee.

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
