# Peer review of "Evaluation of Food Offerings for Workers in Commercial Foodservices from the Perspective of Healthiness and Sustainability"

_ijerph, 2026, doi:10.3390/ijerph23010071_

Round 1

Reviewer 1 Report

Comments and Suggestions for Authors

Summary of the paper

The Authors evaluate menu quality from five commercial restaurants integrating social, health, and environmental sustainability dimensions in Brazil. In the Introduction section, the Authors provide the study background, importance of the topic, and justification of study. Also, they state the research objective and research questions and provide a summary of findings.  In the Materials and Methods section, they state the study area, data collection techniques methods of analysis, and ethical considerations. In the Results section, results are presented and structured consistently with the questionnaire. In the Discussion section, the Authors provide discussion of results – in the context of previous studies –, as well as limitations of study. In the conclusion section, the Authors provide a summary of the findings, conclude based on findings, and concise policy advice.

Overall evaluation and key concerns

The paper is informative. It provides interesting analysis on a key issue in Public Health – Quality Menu provided by Commercial Food Services. The main contribution lies in the dataset allowing the Authors to conclude obesity is influenced by menu composition, social factors and working conditions. Also, the origin and type of protein have a greater environmental influence than its quantity. Given that food quality offered by commercial food services is a subject of continuous public and academic discussions, carrying out this analysis is apt.

However, some important issues must be addressed before the manuscript can be considered for publication. I summarize my concerns in the following points:

Abstract.

  • Clarity of objective.  The entire text of the paper suggest that Authors focus on menu quality and not just “menus”. Consequently, I suggest revising the stated objective to include “quality”.
  • Conclusion. Authors write “Weaknesses in food services related to health, social context, and sustainability highlight the need for systemic, evidence-based, multidimensional policies for healthier, more equitable, and sustainable food environments for workers”. To the best of my understanding, this is not a conclusion, rather it is implication of findings. Authors might want to revise the text clearly stating the conclusion.

Introduction.

  • Paragraphing. The section is poorly paragraphed, making it difficult to follow the flow of ideas. Authors should consider separating the text into paragraphs for distinct messages.
  • Line 88-92. Rather than summarizing the findings here, I suggest highlighting the study’s main contribution, as the text in lines 88–92 is repetitive. A couple of sentences highlighting the study’s main contribution will be sufficient
  • Organization of paper. Authors might want to provide the structure of the paper at the end of this section to give readers a fair idea of the content of subsequent sections. A couple of sentences describing the organization of the paper would be enough.
  • Proofreading required. Please proofread to correct grammatically errors.

Materials and Methods. 

  • Sample size. Based on the Abstract and Results sections, different sample sizes were used (261, 438, and 111). However, these sample sizes are not specified in this section. The authors should clearly state the sample size(s) used in the study, their distribution across the 5 selected units, explain how they were obtained, and clarify why there is variation across the samples. Additionally, the authors should describe the sampling method used, especially given the statement in line 97 that the five units “served approximately 600 employees.” A couple of sentences addressing these issues would be sufficient.

Results.

  • Organization. The organization of this section makes it difficult to follow the presentation of results in relation to the research question (lines 84-87). I suggest restructuring the section to focus explicitly on addressing the research question stated in lines 84–87. This would allow readers to directly link the research question to the corresponding results. Based on the research question, the section could be organized into subsections addressing: (i) the influence of served meals on worker health; (ii) the influence of served meals on food security; and (iii) the influence of served meals on worker engagement with sustainability.
  • • Tables 1 and 2 are hard to follow; consider formatting them to fit (each) on one page each.
  • Proofreading required to correct grammatical errors. 

Discussion.

  • Study limitations. Authors write “[…] this study has limitations, including a convenience sample, restriction to five restaurants in a single city, and exclusive use of BMI, which could be complemented by other anthropometric measures.” While these are true, Authors fail to discuss why they are indeed limitations (what are their implications?). A couple of sentences addressing this will be enough.
  • Future Research.  Authors might want to propose future research topics based on their study limitations. What should be the direction of future research? A couple of sentences addressing this will be enough. 
  • Proofreading required to correct grammatical errors. Please check the use of colons in this section. 

Conclusions 

  • I suggest beginning the section by clearly stating the study objective, to help readers maintain focus on it
  • Proofreading required to correct grammatical errors. Please check the use of colons in this section. 
Comments on the Quality of English Language

Proofreading is required to improve the quality of English.

Author Response

We greatly appreciate your careful reading and corrections.

The responses to the comments are provided in the file ‘Response to Reviewer 1’.

Reviewer 2 Report

Comments and Suggestions for Authors

The manuscript presents a solid, relevant, and well-structured study that integrates social, nutritional, and environmental dimensions of food provision in commercial food services for workers, a population often overlooked in public health research.

One of the main strengths of the study is its multidimensional approach, combining food insecurity indicators, nutritional status, menu quality, and environmental footprints. This integrative framework significantly enhances the analytical depth and provides a systemic understanding of workplace food environments. The data triangulation strategy is clearly described and effectively applied, allowing interpretation beyond simple linear associations.

The Methods section is rigorous and transparent, employing validated tools (EBIA, NOVA, PAHO, environmental footprints), which strengthens the reliability of the findings. The Results are clearly presented, supported by well-designed tables and figures, and the Discussion engages appropriately with current literature, situating the findings within contemporary debates on sustainability, nutrition, and social inequality.

Minor suggestions include:

  • A light stylistic revision of the English to improve flow in some longer paragraphs.

  • Further emphasis on practical implications for food service management in the Conclusions section.

Overall, this is a high-quality manuscript with significant contributions to public health nutrition, collective food services, and sustainable food systems.

Author Response

We greatly appreciate your careful reading and corrections.

The responses to the comments are provided in the file ‘Response to Reviewer 2`.

Reviewer 3 Report

Comments and Suggestions for Authors

This article is a vital result for the food insecurity in Brazil, and how to prevent this problem. however, this article should be revised to improve the quality of this article as this follows;

  1. Introduction: You should explain for more clearly concept for the statement of problems as this view point: Food Insecurity in the global level, problem of Food insecurity in your region, food insecurity in your nation, the problem of food insecurity. Why you study at the workers in commercial food services? Please explain more and add the several of intext-citation in the part of introduction. 
  2. Methodology: Please add the geographic picture to locate your area. And please set the sub-topic to clear concept of your research methodology.
  3. Result: Table 1-2 should provide for the supplementary material. Where is the location of R1 to R5 ? and Why you select only R1 to R5? please explain for more information
  4. Discussion: P-value should display in form <0.05 or >0.05. Delete the year of intext citation according to the format of this journal. Moreover, you should give some of recommendation for the further study.
  5. Reference: Please add more references in English  

Author Response

We greatly appreciate your careful reading and corrections.

The responses to the comments are provided in the file ‘Response to Reviewer 3`. 

Round 2

Reviewer 3 Report

Comments and Suggestions for Authors

All comments from the previous version have been addressed in the current revision.